# Anti-Malignant Ascites Effect of Total Diterpenoids from *Euphorbiae ebracteolatae* Radix Is Attributable to Alterations of Aquaporins via Inhibiting PKC Activity in the Kidney

**DOI:** 10.3390/molecules26040942

**Published:** 2021-02-10

**Authors:** Yuanbin Zhang, Dongfang Liu, Fan Xue, Hongli Yu, Hao Wu, Xiaobing Cui, Xingde Zhang, Hepeng Wang

**Affiliations:** 1School of Pharmacy, Nanjing University of Chinese Medicine, Nanjing 210023, China; 18852000500@163.com (Y.Z.); ldf1512@163.com (D.L.); 13588175351@163.com (F.X.); xiaobingcui@163.com (X.C.); xingde2293@126.com (X.Z.); 13203805708@163.com (H.W.); 2Jiangsu Key Laboratory of Chinese Medicine Processing, Nanjing University of Chinese Medicine, Nanjing 210023, China; 3Engineering Center of State Ministry of Education for Standardization of Chinese Medicine Processing, Nanjing 210023, China; 4State Key Laboratory Cultivation Base for TCM Quality and Efficacy, Nanjing University of Chinese Medicine, Nanjing 210023, China

**Keywords:** *Euphorbia ebracteolata* Hayata, diuresis, bioactive components, terpenoids, kidney cells, PKC inhibitor

## Abstract

This study evaluated the anti-ascites effect of total diterpenoids extracted from *Euphorbiae ebracteolatae* Radix (TDEE) on malignant ascitic mice and elucidated its underlying mechanism. TDEE was extracted by dichloromethane and subjected to column chromatography. The purity of six diterpenoids isolated from TDEE was determined to be 77.18% by HPLC. TDEE (3 and 0.6 g raw herbs/kg, p.o.) reduced ascites and increased urine output. Meanwhile, analysis of tumor cell viability, cycle and apoptosis indicated that TDEE had no antitumor activity. In addition, the expression levels of aquaporins (AQPs) and the membrane translocation levels of protein kinase C (PKC) α and PKCβ in kidney and cells were measured. TDEE reduced the levels of AQP1–4, and inhibited PKCβ expression in membrane fraction. Four main diterpenoids, except compound **2**, reduced AQP1 level in human kidney-2 cells. Compounds **4** and **5** inhibited AQP2–4 expression in murine inner medullary collecting duct cells. The diterpenoid-induced inhibition of AQP1–4 expression was blocked by phorbol-12-myristate-13-acetate (PMA; agonist of PKC). The diterpenoids from TDEE are the main anti-ascites components. The anti-ascites effect of diterpenoids may be associated with alterations in AQPs in the kidneys to promote diuresis. The inhibition of AQP1–4 expression by TDEE is related to the inhibition of PKCβ activation.

## 1. Introduction

*Euphorbia ebracteolata* Hayata (EEH) is a member of the genus *Euphorbia* in the family Euphorbiaceae and is widely used in the treatment of abdominal distension and edema, parasitic infection, tuberculosis, and other diseases in traditional Chinese medicine (TCM) [1]. The root of EEH (EER) contains various terpenoids, which are considered to be effective components. Recent studies on terpenoids have been conducted to determine their anti-inflammatory, antiviral, anti-tuberculosis, and other effects in vitro [2,3,4,5]. However, basic research investigations on the traditional effects of EER in the treatment of abdominal distension and edema are limited. According to theory of TCM, EER can rapidly eliminate abdominal distension and edema by promoting urination and diarrhea [6]. Our previous studies have shown that total diterpenoids from EER (TDEE) can cause diarrhea by regulating the expression of aquaporin (AQP)-1 and AQP3 [7]. However, to date, the effective substances of anti-ascites in EER remain unclear.

Abdominal distension and edema are described as ascites in modern medicine. Ascites is a pathological accumulation of peritoneal fluid [8]. There are many causes of ascites formation, such as cirrhosis, tumor, heart failure, and nephrotic syndrome [9]. The occurrence of ascites is related to portal hypertension, which leads to splanchnic arterial vasodilation, reduction in the effective circulating volume, activation of endogenous vasoconstrictor systems, and avid water retention in the kidneys [10]. AQPs are rapid membrane transporters of water for crossing epithelial and endothelial barriers and thus play a fundamental role in maintaining water balance [11]. Numerous studies have shown that AQPs play a key role in the formation and disorder of ascites [12,13,14,15]. There are many kinds of AQPs in the kidneys, such as AQP1, AQP2, AQP3 and AQP4 [16]. Because TDEE can regulate the expression of AQPs in the intestine [7], we propose that TDEE may be utilized in the treatment of ascites by regulating AQPs in the kidneys.

AQPs play a critical role in urine formation, however the mechanisms of regulating AQPs expression in kidneys have not been clarified. Currently, only a limited number of reports indicate that AQPs are regulated through protein kinase C (PKC) and protein kinase A (PKA) pathways [17,18,19,20]. Some terpenoids isolated from *Euphorbia* (e.g., euphol and ingenol 3-angelate) have regulatory effects on PKC. TDEE may regulates AQPs expression in kidneys by PKC pathway.

In this study, we extracted TDEE and evaluated its anti-ascites effect as well as analyzed AQPs expression, PKC and PKA activities in the kidneys of ascitic mice. Then, we isolated and purified the main diterpenoids from TDEE to investigate its effects on AQPs protein and mRNA expression levels in two kinds of kidney cells (human kidney-2 cells (HK-2) and murine inner medullary collecting duct cells (mIMCD3)). Finally, we choose a diterpenoid compound (euphebracteolatin A), using phorbol-12-myristate-13-acetate (PMA; agonist of PKC) to activate PKC pathway, to identify whether the expression of AQPs in kidneys is regulated by PKC pathway. This study aimed to determine the anti-ascites effect of TDEE in EER and provide a basis for the clinical application of EER and identification of candidate drugs for diuresis and anti-ascitic effects.

## 2. Results

### 2.1. Effects of TDEE on Ascitic Fluid Weight, Urine Weight, and Fecal Water Content

Treatment of ascitic mice with total extract of EER (TEER) and TDEE (3 and 0.6 g raw herbs/kg, p.o.) and furosemide (positive drug) significantly reduced the increase in ascitic fluid weight compared to the model group (Figure 1A). Figure 1C showed that the urine weights significantly (*p* < 0.05, compared to model group) increased in ascitic mice pretreated with TEER, TDEE, and furosemide. In addition, in the TEER and TDEE groups at dose of 3 g raw herbs/kg, a significant increase in fecal water content was observed compared with the model group (Figure 1D). However, the non-diterpenoid fraction of EER (NTDEE) groups showed no significant influence on ascites weight, urine weight, and fecal water content compared to the model group.

### 2.2. Effects of TDEE on Serum Albumin Levels in Ascitic Mice

Serum albumin is the most abundant protein in vertebrate plasma and can maintain the osmotic pressure of blood and regulate water transport. Figure 1B showed that the production of ascites in model mice caused a significant decrease in serum albumin compared to the normal group. After treatment with furosemide, TEER and TDEE (3 and 0.6 g raw herbs/kg, p.o.), the levels of serum albumin were effectively recovered compared to the model group. No improvement in this index was observed in the NTDEE groups.

### 2.3. Effects of TDEE on Tumor Cell Viability, Cycle, and Apoptosis in Ascitic Mice

In malignant ascites, tumor cells float in the ascitic fluid. We evaluated whether the reduction in ascites is related to the anticancer activity of the TEER and TDEE. The results showed that these drugs did not inhibit the tumor cell viability (Figure 2A) and did not lead to cell cycle arrest (Figure 2B) and apoptosis (Figure 2C,D).

### 2.4. Effects of TDEE on AQPs Expression in the Kidneys of Ascitic Mice

The kidney is an important organ for water reabsorption and urinary concentration. AQP water channels mediate rapid transport of water in the kidneys. Thus, we examined the expression of AQPs in the kidneys. Figure 3 showed that the enhancement of ascites had little effects on AQPs expression in the kidneys in comparison with the normal group. When treated with TEER and TDEE, the expression of AQP1, AQP2, AQP3 and AQP4 proteins was markedly lower than in the model group at most doses. TEER resulted in a significant decrease in AQP2, AQP3 and AQP4 expression at doses of 3.0 and 0.6 g raw herbs/kg, whereas AQP1 expression was only reduced at a low dose (0.6 g raw herbs/kg). Notably, TDEE had the same inhibitory effects on AQP3 and AQP4 expression as TEER at two doses. The difference between TDEE and TEER is that TDEE inhibited the expression of AQP1 and AQP2 only at high dose (3 g raw herbs/kg). NTDEE had weak effects on AQPs levels in kidneys, only regulated the expression of AQP4 protein at a dose of 3 g raw herbs/kg.

### 2.5. Effects of TDEE on PKC and PKA Activation in the Kidneys of Ascitic Mice

Previous studies have indicated that AQPs expression could be regulated by PKC and PKA activation. We examined the effects of TDEE on PKC-α, PKC-β and PKA activation. PKC activation was measured by its translocation from the cytosol to cell membrane. As shown in Figure 4A,B, the protein expression level of PKC-β in the membrane fraction was significantly decreased in TDEE groups, whereas there was no significant effect on the expression level of PKC-α membrane protein. Phospholipase C (PLC) is an important upstream signaling molecule that mediates the activation of PKC. TDEE treatment did not have any effect on the levels of phosphorylated and total PLC (Figure 4D). The PKA is a heterotetramer consisting of two catalytic subunits bound to two regulatory subunits. The release of catalytic subunits is a hallmark of PKA activation. In Figure 4C, TDEE did not affect the expression of PKA catalytic subunits.

### 2.6. Determination of Content of Six Diterpenoids in TDEE by High-Performance Liquid Chromatography (HPLC)

The results of animal experiments showed that TDEE was the main effective fraction of EER. We isolated six diterpenoids (the structures of compounds **1** to **6** are shown in Figure 5; the spectral data were added into the Appendix A) from TDEE and used these compounds as standards to determine diterpenoids content in TDEE. The HPLC chromatograms for TDEE and the six diterpenoid standards are presented in Figure 6. The content of compounds **1** to **6** were 1.07%, 12.00%, 12.69%, 9.15%, 39.26% and 3.02%, respectively, accounting for 77.18% of TDEE.

### 2.7. Effects of Diterpenoids from TDEE on AQPs Expression in Kidney Cells

TDEE was the active fraction of EER that inhibited ascites by reducing AQP1, AQP2, AQP3 and AQP4 expression in the kidneys. At the same time, TDEE also significantly reduced the protein and mRNA expression of AQP1, AQP2, AQP3 and AQP4 in human kidney-2 cells (HK-2) and murine inner medullary collecting duct cells (mIMCD3) (Figure 7). The main components of TDEE are diterpenoids. We compared the effects between TDEE and diterpenoids mixture (six isolated diterpenoids were mixed according to their content in TDEE). The results showed that the inhibitory effects of diterpenoids mixture and TDEE on AQP1, AQP2, AQP3 and AQP4 expression were similar (Figure 7). The activities of four main diterpenoids from TDEE were evaluated in vitro. Figure 8 and Figure 9 showed that the protein and mRNA expression levels of AQP1 in HK-2 cells were significantly reduced by compounds **3**, **4** and **5**. Treatment with compound **2** did not induce a notable decrease in AQP1 at the protein and mRNA levels. For these compounds, compound **3** (an ent-abietane type diterpenoid) was more active than the other compounds. In mIMCD3 cells, four diterpenoid compounds induced different degrees of inhibition of AQP2, AQP3 and AQP4 expression. The results of the protein and gene expression are presented in Figure 8 and Figure 9. Compounds **4** and **5** possessed potent AQP2, AQP3 and AQP4 inhibitory effects compared with the control group at the gene and protein levels. Compounds **2** and **3** only showed different effects on *Aqp2*, *Aqp3* and *Aqp4* mRNA expression.

### 2.8. Effects of Activator of PKC on Diterpenoid-Induced Inhibition of AQPs

The results from the animal experiment indicated that TDEE simultaneously inhibited PKCβ activity and AQPs expression in the kidneys of H22 tumor cell-bearing mice. To further clarify whether the PKC regulated the expression of AQPs in kidney cells, the PKC agonist (PMA) was pretreated in HK-2 and mIMCD3 cells at 1.5 h before diterpenoid (Euphebracteolatin A) stimulation. The change of membrane PKCβ, AQP1, AQP2, AQP3 and AQP4 expression was detected. Western Bolt results showed that euphebracteolatin A could reduce the protein expression of membrane PKCβ, AQP1, AQP2, AQP3 and AQP4, and the effects could be attenuated by PMA pretreatment (Figure 10).

## 3. Discussion

EER has historically been used in TCM for the treatment of abdominal distension and edema [1]. In this study, the anti- ascites effects of EER were investigated using H22 tumor cell-bearing ascitic mice. Malignant ascitic mice are widely used in research to study tumor and ascites treatments [21,22]. Our results showed that both EER and TDEE can significantly reduce ascites volume, whereas NTDEE had no effect. These results indicated that TDEE was the anti-ascites active fraction of EER. Previous studies have shown that the diterpenoids from EER have inhibitory activities on tumor cells in vitro [5,23]. However, this study found that they have no effects on the tumor cell viability, cell cycle, or apoptosis of tumor cells. Possible reasons for this discrepancy may include the differences in doses and poor correlation between in vivo and in vitro activities. TCM holds the view that EER is effective in the treatment of edema mainly by rapidly regulating the excretion of urine and feces [6]. Therefore, we analyzed the effect of TDEE on urine weight and fecal water content in ascitic mice. It should be noted that urine weight and ascitic fluid weight were both influenced by TDEE at doses of 3 and 0.6 g raw herbs/kg, while fecal water content only increased at 3 g raw herbs/kg, so TDEE mainly decreased ascites volume by reducing reabsorption of water in the kidneys. Renal water reabsorption plays a central role in the maintenance of body water homeostasis [16]. AQPs provide the main channels for the kidneys to reabsorb water and concentrate urine [24]. Consequently, the study of AQPs expression may provide a mechanism for decreasing ascites.

Various AQPs are expressed in different areas of the kidney. AQP1, AQP2, AQP3 and AQP4 are localized in renal epithelial cells and are closely related to water reabsorption in the kidney [25]. AQP1 is highly expressed in both apical and basolateral membranes of the proximal tubules and control >70% of water absorption in the glomerular filtrate [16]. AQP2, AQP3 and AQP4 are exclusively expressed in the collecting ducts, which are vital segments for regulating water balance and urine concentration [16]. In the present study, TDEE reduced the protein expression of AQP1, AQP2, AQP3 and AQP4 in mice kidneys. These findings were confirmed by the results of cell experiments in vitro. The AQPs inhibitory activities of the main diterpenoids in TDEE were assessed in the cell lines of the proximal tubule and collecting duct. Our results showed that diterpenoids in TDEE reduced AQPs protein and mRNA expression levels in HK-2 and mIMCD3 cells.

The relative expression levels of AQPs protein and mRNA of each diterpenoid were compared with the control group. Renal tubules are an important component of water reabsorption. Among the four diterpenoids, ent-abietane diterpenoids jolkinolide B (12.69% of TDEE) had the strongest effect on AQP1 regulation in HK-2 cells. By comparing it with the structure of compound **2**, jolkinolide B (compound **3**) has two epoxy rings at position of C-11 and 14. When the epoxy ring at C-11 and 14 were, respectively, converted to a hydroxyl and double bond in compound **2**, the inhibitory activity of AQP1 disappeared. Urine concentration mainly occurs in the renal collecting tubules. The results from the mIMCD3 cells showed that the rosane diterpenoid compound **4** and norditerpenoid lactone compound **5** (accounting for 9.16% and 39.26% of TDEE, respectively) have the strongest inhibitory effects on AQP2, AQP3 and AQP4 expression. The inhibitory effects of ent-abietane diterpenoids were weak compared with these two types of diterpenoids. The inhibitory effects of different type of diterpenoids on various AQPs in the kidney were not the same, which may be due to variations in the chemical structure of diterpenoids and response time of AQPs to each compound. Overall, these diterpenoids may impart synergistic inhibitory effects of AQPs. Because the activities of those diterpenoids were estimated in vitro, the activities of those compounds after metabolism in vivo remain unclear and thus need further investigation.

Several studies have suggested that PKC and PKA function in the regulation of AQPs [17,18,19,20]. PKC is a family of serine/threonine kinases involved in various cellular signal transductions [26]. The PKC family has included several isoforms [26]. After activation, the PKCs translocate from the cytosol to the plasma membrane, bind to diacylglycerol (DAG), and subsequently induce a series of physiological changes [27]. PKC translocation from the cytosol to the plasma membrane is an indicator of PKC activation [28]. DAG is generated by the PLC catalyzed hydrolysis of membrane phosphatidylinositol-4,5-bisphosphate (PIP2) [27]. In animal experiments, we measured the expression levels of PKCα and PKCβ in membrane fraction. At the same time, we also detected the changes in the upstream proteins of PKC. The results suggested that TDEE decreased the expression of PKCβ on the cell membrane and inhibited the activity of PKCβ. However, no effect of TDEE on phosphorylation level of PLC was observed. The TDEE may directly bind PKCβ and inhibit the activation of PKCβ. PKA enzymatic activity is primarily controlled by regulatory subunits that form a holoenzyme complex with the catalytic subunits [29]. The release of catalytic subunits is a hallmark of PKA activation [29]. We found that TDEE treatment had no effect on PKA activation. To explore whether the inhibition of AQP1, AQP2, AQP3 and AQP4 in kidneys is mediated through blocking the activation of PKC signals, we used PMA as PKC agonist to interfere with the expression of AQP1, AQP2, AQP3 and AQP4 in HK-2 and mIMCD3 cells. Our results showed that euphebracteolatin A (the diterpenoid from TDEE) decreased the expression levels of AQP1, AQP2, AQP3 and AQP4 in kidney cells. Euphebracteolatin A-induced inhibition of AQP1, AQP2, AQP3 and AQP4 expression was blocked by PMA. Thus, we infer that TDEE regulated the expression of AQP1, AQP2, AQP3 and AQP4 by inhibiting the activation of PKCβ. PKC isoforms are numerous; TDEE may also regulate AQPs expression by inhibiting other isoforms that await further research. To date, the molecules downstream of PKC that regulate AQPs expression remain poorly understood. In the next step, we will focus on the downstream regulatory mechanisms of PKC.

In summary, TDEE is an active fraction of EER against ascites. TDEE consists of diterpenoids such as ent-abietane, rosane and norditerpene lactones types. TDEE and its diterpenoids can inhibit the reabsorption and concentration of water in the kidneys by reducing the expression levels of AQP1, AQP2, AQP3 and AQP4, thereby resolving ascites. The inhibition of AQP1, AQP2, AQP3 and AQP4 expression in kidneys by TDEE is related to the inhibition of PKCβ activation. This study may provide the basis for the clinical application of EER and the identification of candidate drugs that impart diuresis and anti-ascites effects.

## 4. Materials and Methods

### 4.1. Chemicals and Reagents

An albumin assay kit was purchased from Nanjing Jiancheng Bioengineering Institute (Nanjing, China). HPLC-grade acetonitrile was purchased from Tedia Co. (Anqing, China). Dulbecco’s modified Eagle’s medium (DMEM) was obtained from HyClone (Logan, UT, USA). Minimum essential medium with non-essential amino acids (MEM NEAA) and fetal bovine serum (FBS) were obtained from Procell Life Science & Technology Co., Ltd. (Wuhan, China). Rabbit-anti-AQP1 and AQP3 antibody were obtained from EnoGene Co., Ltd. (Nanjing, China). Rabbit-anti-AQP2, AQP4, and mouse-anti-β-actin antibodies were purchased from Zen BioScience Co., Ltd. (Chengdu, China). Rabbit-anti-PKCα, PKCβ, PLC and phospho-PLC (Tyr783) antibodies were purchased from Affinity Biosciences Co., Ltd. (Changzhou, China). Radio immunoprecipitation assay buffer (RIPA buffer), cell cycle and cell apoptosis detection kits were provided by Yifeixue Biotechnology (Nanjing, China). A BCA kit was purchased from Beyotime Biotechnology (Shanghai, China). TRIzol reagent buffer were purchased from Thermo Fisher Scientific (Carlsbad, CA, USA). PVDF membranes were purchased from Millipore (Bedford, MA, USA). A cDNA synthesis kit was purchased from YEASEN Biotech Co., Ltd. (Shanghai, China).

### 4.2. Plant Material

The EER, the root of *Euphorbia ebracteolata* Hayata, was purchased from Baohetang Pharmaceutical Co., Ltd. (Bozhou, China) on September 15, 2017 and authenticated by Professor Haobin Hu of the Jiangsu Institute for Food and Drug Control. A voucher specimen (WH20170915) was deposited at the Herbarium of Traditional Chinese Medicine at the School of Pharmacy of Nanjing University of Chinese Medicine.

### 4.3. Extraction and Isolation of TDEE and Purification of Diterpenoids

The EER were pulverized using a mechanical grinder. The powder (100 g) was subjected to successive extraction on a Soxhlet extractor for 6 h with dichloromethane. After that, the extract was concentrated under reduced pressure, which yielded dichloromethane extract. The dichloromethane extract was subjected to column chromatography (CC) on ODS eluting with 40% and 90% methanol sequentially. The eluent of 90% methanol was collected, concentrated and dried to prepare the TDEE (2.4 g). The extracted residues were merged and extracted thrice with 95% ethanol. The extract was then combined and evaporated to dryness to obtain NTDEE (15.6 g). TEER (19.8 g) was extracted from the EER powder (100 g) with 95% ethanol. These extracts were used in the animal experiments.

The main compounds of TDEE were isolated using the following process. The root of EER (20 kg) was powdered and extracted with dichloromethane over 72 h at room temperature by maceration thrice. The dichloromethane extract was obtained and concentrated using a rotary evaporator. The residue was subjected to CC on silica gel using a gradient system of increasing polarity with CH_2_Cl_2_-MeOH (100:0–0:100) to afford 8 fractions (Fr. 1–Fr. 8). Fr. 4 was subjected to a silica gel column eluting with petroleum ether-EtOAc (from 50:1 to 1:1), followed by reversed-phase ODS CC eluting with MeOH-H_2_O from 60% to 80% to yield compounds **5** and **6**. Fr. 5 was chromatographed on a silica gel column eluting with petroleum ether-EtOAc (from 35:1 to 0:1) to yield 7 subfractions (Fr. 5. 1–Fr. 5. 7). Fr. 5. 3 was separated by ODS CC eluting with a MeOH-H_2_O (from 75% to 100%), followed by semi-preparative HPLC (MeOH-H_2_O, from 75% to 100%) to afford compound **4**. Fr. 5.4 was chromatographed on a silica gel column eluting with petroleum ether-EtOAc (from 30:1 to 5:1), followed by a Sephadex LH-20 (MeOH:CH_2_Cl_2_, 1:1) and semi-preparative HPLC (MeOH-H_2_O, from 60% to 95%) to afford compound **3**. Fr. 5.6 was purified by ODS CC eluting with a MeOH-H_2_O gradient from 40:60 to 100:0, followed by Sephadex LH-20 CC (CH_2_Cl_2_-MeOH, 1:1) to yield compounds **1** and **2**.

### 4.4. In Vivo Experiments

#### 4.4.1. Animals and Experimental Design

ICR male mice (weight: 18–22 g) were obtained from the Qing Long Shan Animal Breeding Farm (Nanjing, China). The mice were maintained in standard laboratory cages in moderate humidity (50% ± 5%) at constant temperature (22 ± 1 °C) in a 12 h light–dark cycle room. All animals had free access to food and water during the experimental period. The experimental protocol was approved by the Animal Ethics Committee of Nanjing University of Chinese Medicine (Nanjing, China).

A total of 90 male mice were randomly divided into 9 groups (*n* = 10). The control group was injected intraperitoneally with 0.2 mL saline. The mice in the other groups were i.p. inoculated with 1 × 10^6^ H22 cells in a volume of 0.2 mL. All the drugs were suspended in water with 0.5% (*w*/*v*) sodium carboxyl methyl cellulose (0.5% Na-CMC). After 24 h, mice were treated via gavage every day according to the following scheme for 8 days: normal and model groups were administered 0.5% Na-CMC. Furosemide (6 mg/kg) was chosen as positive control because it is extensively used to treat edematous diseases such as ascites by diuresis effect. The TEER, TDEE, and NTDEE groups were administered corresponding extracts of EER (3.0 and 0.6 g raw herbs/kg).

#### 4.4.2. Ascitic Fluid Weight, Urine Weight, and Fecal Water Content Assays

The mice from the model and treated groups were sacrificed, and ascitic fluid was collected from the peritoneal cavity. The fluid was then weighed on an electronic balance.

The mice were housed in individual cages. The feces were collected into weighed and sealed vials. After 5 h, the vials containing moist feces were weighed. Then the vials were dried to constant weight at 105 °C. The fecal water content (%) = ((weight of vials containing moist feces − weight of vials containing dried feces)/(weight of vials containing moist feces − weight of vials)) × 100.

The weight of urine excretion was measured by modification of a previous method [30]. The mice were housed in individual beakers. The beakers were covered with a metal net on the mouths, then inverted on the plastic dishes containing superabsorbent polymer (SAP). In this test, the feces were cleaned. After 5 h, the plastic dishes containing SAP and urine were weighed. The following equation was used to determine urine weight: urine weight (g) = weight of the plastic dishes containing SAP and urine − weight of the plastic dishes containing SAP.

#### 4.4.3. Serum Albumin Analysis

Blood from all groups was collected by orbital puncture and centrifuged at 5000× *g* for 10 min at 4 °C. Serum albumin was measured using an albumin assay kit.

#### 4.4.4. Tumor Cell Viability, Cycle, and Apoptosis Assays

After the mice were sacrificed, the ascitic fluid containing the tumor cells was collected from the peritoneal cavity and diluted 20 times with saline immediately. The viability of tumor cells was assessed by 0.4% trypan blue staining and counted in an automated cell counter (ALIT Life Science Co., Ltd., Shanghai, China).

The diluted cells suspension (0.5 mL) was centrifuged at 1000 g for 5 min at 4 °C. The cells were collected and fixed in cold 70% ethanol at 4 °C overnight. After fixation, tumor cells were washed twice in PBS and centrifuged, and then incubated with RNase A solution (100 μL) for 30 min at 37 °C. Finally, cells were incubated with 400 μL propidium iodide (PI) for 30 min at 4 °C in the dark and filtered through a 300-mesh nylon membrane. The percentage of cells in different phases was analyzed using an Accuri C6 Flow Cytometer (BD Biosciences, San Jose, CA, USA).

Tumor cell apoptosis was detected using an Annexin V-FITC/PI Apoptosis kit. The diluted cells suspension (0.5 mL) was washed twice in PBS and centrifuged at 1000 g for 5 min at 4 °C. The cells were collected and resuspended in 100  uL binding buffer followed by incubation with 5 μL Annexin V-FITC and 10 μL PI staining solution for 15 min at room temperature in the dark. Following incubation, 400 μL binding buffer was added to each tube. The cell apoptosis was detected by using an Accuri C6 Flow Cytometer.

#### 4.4.5. Western Blot Analysis of AQPs, PKC, PKA and PLC in the Kidneys

After the mice were sacrificed, the kidneys were isolated and snap-frozen and then stored at −80 °C for protein expression analysis. The tissue samples were homogenized in RIPA lysis buffer. After storage at −20 °C overnight, the tissue lysates were collected after centrifugation for 10 min at 12,000× *g*. After centrifugation, the supernatant was collected for the determination of total and phosphorylated proteins. The membrane proteins were extracted from the kidneys using a Membrane Protein Extraction Kit according to the manufacturer’s protocol. The protein concentrations were determined using a BCA kit. The protein solutions were denatured in boiling water for 5 min and mixed with 5× loading buffer. Equal amounts of proteins were separated on 10% SDS-polyacrylamide gel electrophoresis, transferred to polyvinylidene difluoride (PVDF) membranes, and subsequently blocked by 5% albumin from bovine serum (BSA) for 2 h. After blocking, the membranes were incubated with primary antibodies against AQP1 (1:2000), AQP2 (1:1000), AQP3 (1:2000), AQP4 (1:1000), PKCα (1:2000), PKCβ (1:2000), PKA (1:2000), PLC (1:2000), p-PLC (1:800) and β-actin (1:5000) at 4 °C overnight. Then, the membranes were washed with Tris-buffered saline containing 0.1% Tween-20 (TBST), and the primary antibodies were detected with horseradish peroxidase-conjugated secondary antibodies (1:5000). The proteins were visualized with chemiluminescence reagent and a Tanon 5200 chemiluminescence imaging system (Tanon, Shanghai, China). The blot bands were quantified by densitometry using ImageJ.

### 4.5. Determination of Six Major Diterpenoids in TDEE

Euphorin G, ent-11α-hydroxyabicta-8(14),13(15)-dien-16,12-olide, jolkinolide B, euphebracteolatin A, fischeria A, and jolkinolide E were dissolved in acetonitrile into the appropriate concentrations. TDEE was also dissolved in acetonitrile to detect the ingredient contents using HPLC. HPLC analysis was performed on a Waters 2695 separations system equipped with a photodiode array detector (Waters, Milford, CT, USA) and an Agilent Eclipse XDB-C18 column (4.6 mm × 250 mm, 5 μm). A mobile phase consisted of 0.1% formic acid water (A) and acetonitrile (B). The solvent was developed with a flow rate at 1 mL/min using the following gradient program: 0–25 min, 65% B; 25–40 min, 65%–85% B; 40–47 min, 85%–100% B. The injection volume was 10 μL. The column temperature was 30 °C, and a 265 mm wavelength was employed.

### 4.6. In Vitro Experiments

#### 4.6.1. Cell Culture

The HK-2 and mIMCD3 cells were purchased from Procell Life Science & Technology Co., Ltd. (Wuhan, China). All cells were cultured in an incubator in a 5% CO_2_ atmosphere at 37 °C until 90% confluency and then treated with different compounds and managed for the experiments. The HK-2 cells were grown in MEM NEAA supplemented with 10% FBS and 1% antibiotics, whereas the mIMCD3 cells were grown in DMEM medium containing 10% FBS and 1% antibiotics.

#### 4.6.2. Western Blot and Quantitative Reverse Transcription-PCR (qRT-PCR) Analyses of AQPs

HK-2 and mIMCD3 cells were seeded (5 × 10^5^/well) in 6-well plates for 24 and 12 h before treatment. The cells were treated with different compounds for 8 h in 5% CO_2_ incubator at 37 °C. Cells treated with 0.1% dimethyl sulfoxide were used as vehicle control. Post-treatment, cells were washed twice in cold PBS, harvested by scraping in RIPA lysis buffer, and then lysed. After storage at −20 °C overnight, the lysates were clarified by centrifugation at 12,000× *g* for 10 min, the supernatants were collected and used in Western blot analysis. Western blot was conducted as previously described in the animal experiments in this article.

For the isolation of total RNA. After incubation with different compounds for 8 h, the cells were washed by PBS and dissolved in TRIzol reagent buffer, and RNA was isolated according to the manufacturer’s guidelines. cDNA was synthesized from an equal amount of RNA using a cDNA synthesis kit. The reaction mixture for qRT-PCR was prepared by the addition of cDNA, forward and reverses primers mentioned in Table 1, DEPC water, and SYBR green master mix. An Applied Biosystems 7500 Real-Time PCR instrument was used for gene expression analysis. The PCR conditions were as follows: an initial denaturation at 95 °C for 5 min followed by 40 cycles of 95 °C for 10 s and 60 °C for 34 s. Finally, PCR data were analyzed by double delta CT methods on Microsoft Excel spreadsheet.

#### 4.6.3. Effect of PKC agonist PMA on the AQPs Protein Expression in Kidney Cells

The HK-2 and mIMCD3 cells were plated onto 6-well plates at a density of 5 × 10^5^ cells/well. After cells attachment, the cells were pre-incubated with or without 200 nM PKC agonist (PMA) for 1.5 h, and then treated with euphebracteolatin A (2.5 μM) for 8 h. Total and membrane proteins extraction and Western blotting analysis were performed as described in detail previously.

### 4.7. Statistical Analysis

SPSS 19.0 analysis software was used for statistical analysis. The data were analyzed using one-way ANOVA or a rank sum test. Numerical data were expressed as the mean ± standard deviation (SD). Differences with *p* < 0.05 were considered statistically significant.

## Figures and Tables

**Figure 1 molecules-26-00942-f001:**
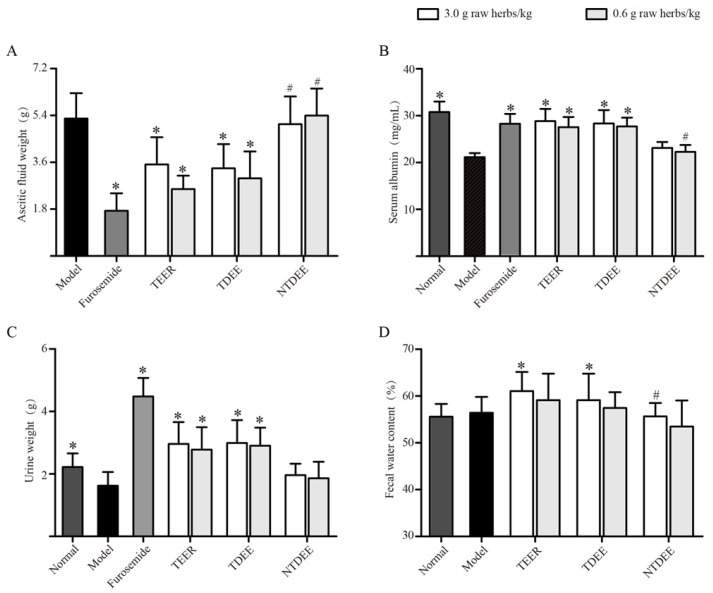
Effects of extracts from the root of *Euphorbia ebracteolata* Hayata (EER) on ascitic fluid weight (**A**), serum albumin levels (**B**), urine weight (**C**), and fecal water content (**D**) in H22 tumor cell-bearing mice. Normal: normal mice treated with blank solvent. Model: H22 tumor cell-bearing mice treated with blank solvent. TEER: total extract of EER; TDEE: total diterpenoids from EER; NTDEE: non-diterpenoid fraction of EER. The data are represented as mean ± SD, *n* = 10, * *p* < 0.05 vs. model group (one-way ANOVA followed by Dunnett test); ^#^
*p* < 0.05 vs. TEER group using the same dose (one-way ANOVA followed by Dunnett test).

**Figure 2 molecules-26-00942-f002:**
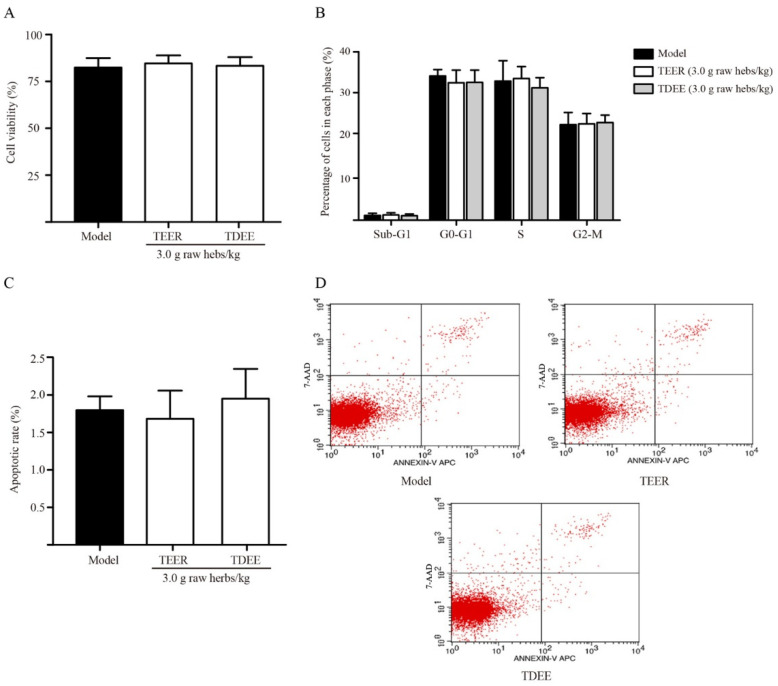
Effects of extracts from the root of *Euphorbia ebracteolata* Hayata (EER) on cell viability, cell cycle and apoptosis in H22 tumor cell-bearing mice. (**A**) Cell viability rate. (**B**) Percentage of cells in each cycle phase. (**C**,**D**) Cell apoptosis with Annexin V-FITC/PI dual-staining (flow cytometry). Model: H22 tumor cell-bearing mice treated with blank solvent. TEER: total extract of EER; TDEE: total diterpenoids from EER. The data are represented as the mean ± SD, *n* = 6. Significant differences compared to model group were analysed by one-way ANOVA with Dunnett test, but there were no significant differences.

**Figure 3 molecules-26-00942-f003:**
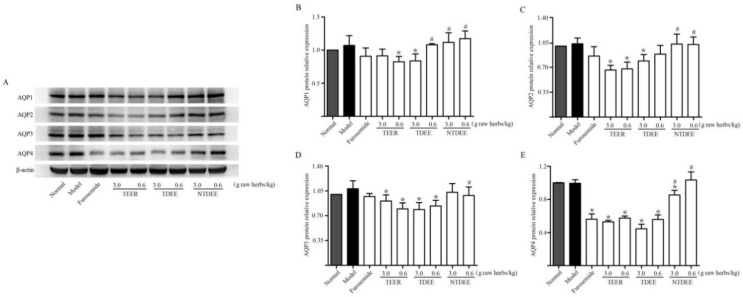
Effects of extracts from the root of *Euphorbia ebracteolata* Hayata (EER) on aquaporins (AQPs) expression in the kidneys of H22 tumor cell-bearing mice. (**A**) Western blot bands of AQP1, AQP2, AQP3 and AQP4. (**B**) AQP1 protein relative expression. (**C**) AQP2 protein relative expression. (**D**) AQP3 protein relative expression. (**E**) AQP4 protein relative expression. Normal: normal mice treated with blank solvent. Model: H22 tumor cell-bearing mice treated with blank solvent. TEER: total extract of EER; TDEE: total diterpenoids from EER; NTDEE: non-diterpenoid fraction of EER. The data are represented as the mean ± SD, *n* = 3, * *p* < 0.05 vs. model group (Mann–Whitney U test); ^#^
*p* < 0.05 vs. TEER group at the same dose (Mann–Whitney U test).

**Figure 4 molecules-26-00942-f004:**
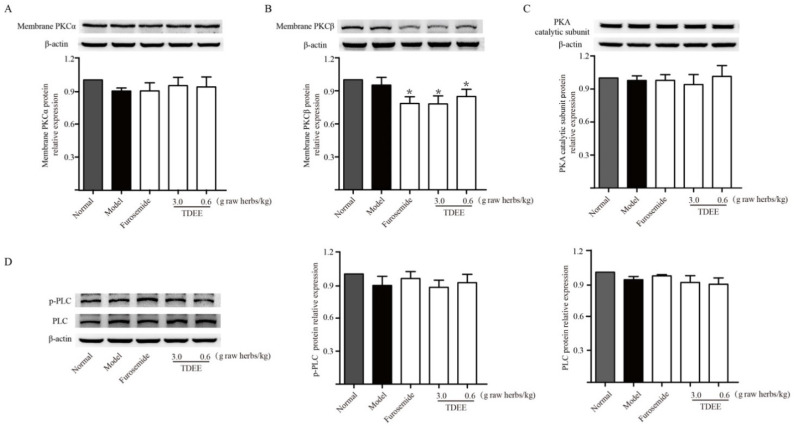
Effects of total diterpenoids from the root of *Euphorbia ebracteolata* Hayata (TDEE) on protein kinase C (PKC) and protein kinase A (PKA) activation in the kidneys of H22 tumor cell-bearing mice. (**A**) Western blot band and protein relative expression of PKCα. (**B**) Western blot band and protein relative expression of PKCβ. (**C**) Western blot band and protein relative expression of PKA. (**D**) Western blot bands and protein relative expression of phospholipase C (PLC) and phosphorylated PLC. Normal: normal mice treated with blank solvent. Model: H22 tumor cell-bearing mice treated with blank solvent. The data are represented as the mean ± SD, *n* = 3, * *p* < 0.05 vs. model group (Mann–Whitney U test).

**Figure 5 molecules-26-00942-f005:**
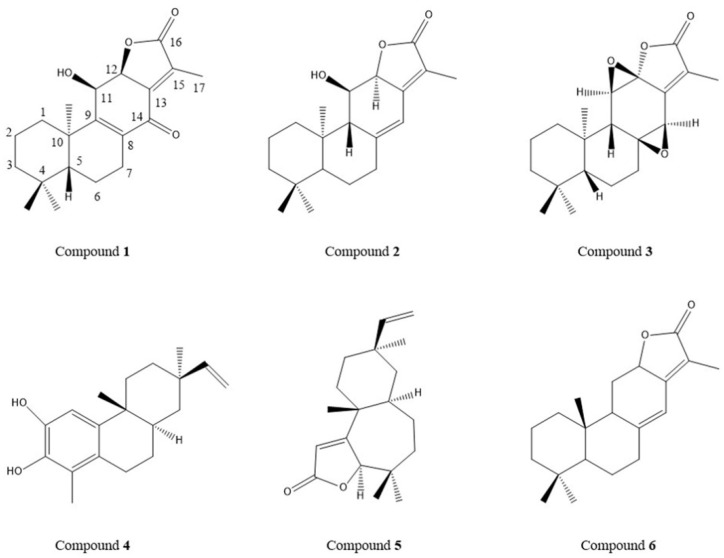
The chemical structures of six diterpenoid compounds. Compound **1**: euphorin G; compound **2**: ent-11α-hydroxyabicta-8(14),13(15)-dien-16,12-olide; compound **3**: jolkinolide B; compound **4**: euphebracteolatin A; compound **5**: fischeria A; compound **6**: jolkinolide E.

**Figure 6 molecules-26-00942-f006:**
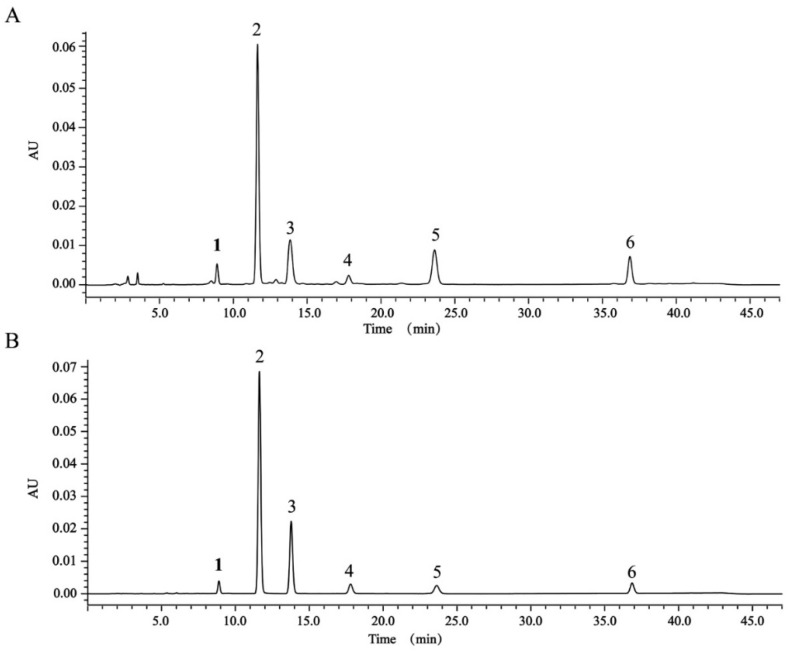
HPLC chromatograms of total diterpenoids from the root of *Euphorbia ebracteolata* Hayata (TDEE) (**A**) and six diterpenoid standards (**B**). Peak 1: euphorin G; peak 2: ent-11α-hydroxyabicta-8(14), 13(15)-dien-16, 12-olide; peak 3: jolkinolide B; peak 4: euphebracteolatin A; peak 5: fischeria A; peak 6: jolkinolide E.

**Figure 7 molecules-26-00942-f007:**
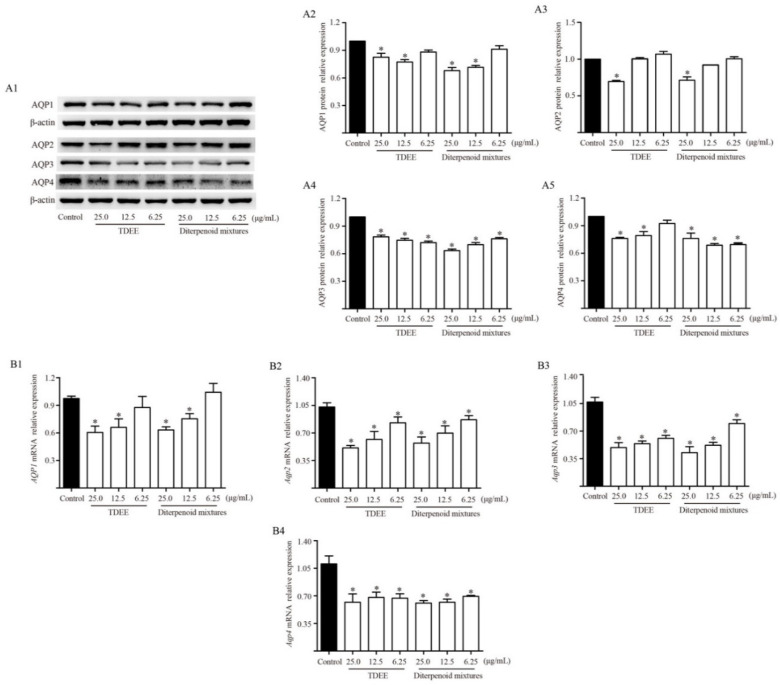
Effects of total diterpenoids from the root of *Euphorbia ebracteolata* Hayata (TDEE) and six diterpenoids mixtures on the protein and mRNA expression levels of aquaporin (AQP)-1 in human kidney-2 cells (HK-2), and AQP2, AQP3 and AQP4 in murine inner medullary collecting duct cells (mIMCD3). (**A1**–**A5**) Western blot bands and protein relative expression of AQP1, AQP2, AQP3 and AQP4. (**B1**–**B4**) mRNA relative expression of *AQP1*, *Aqp2*, *Aqp3* and *Aqp4*. The data are represented as the mean ± SD, *n* = 3, * *p* < 0.05 vs. control group (Mann–Whitney U test). Diterpenoids mixture was mixed according to the contents of six diterpenoids in TDEE.

**Figure 8 molecules-26-00942-f008:**
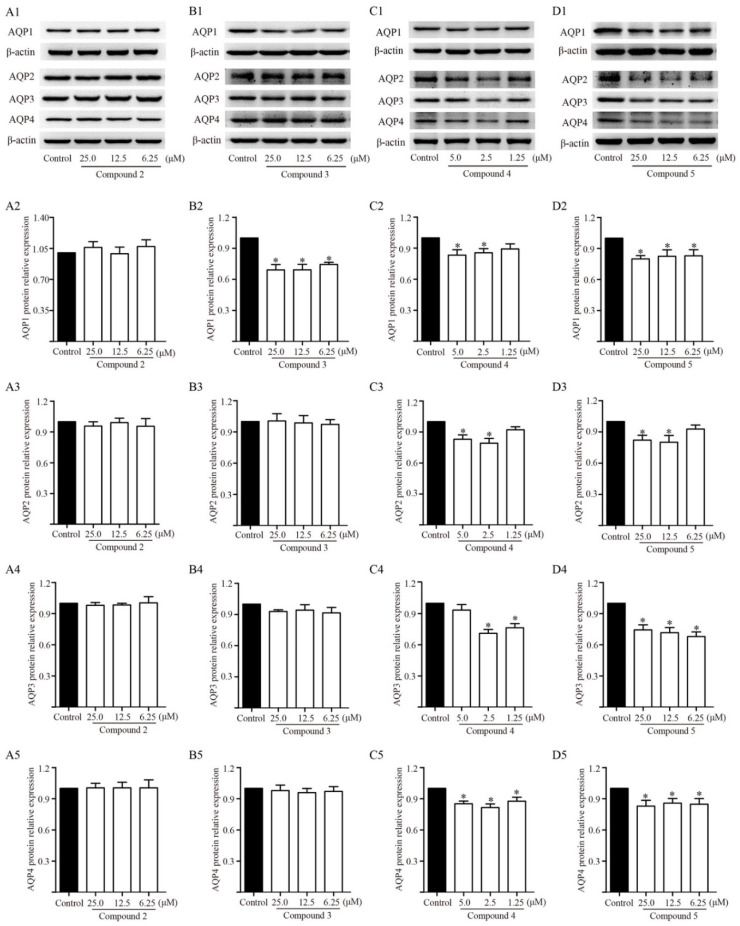
Effects of compounds **2**, **3**, 4 and **5** on protein expression levels of aquaporin (AQP)-1 in human kidney-2 cells (HK-2), and AQP2, AQP3 and AQP4 in murine inner medullary collecting duct cells (mIMCD3). (**A1**–**A5**) Western blot bands and protein relative expression of AQP1, AQP2, AQP3 and AQP4 after compound **2** administration. (**B1**–**B5**) Western blot bands and protein relative expression of AQP1, AQP2, AQP3 and AQP4 after compound **3** administration. (**C1**–**C5**) Western blot bands and protein relative expression of AQP1, AQP2, AQP3 and AQP4 after compound **4** administration. (**D1**–**D5**) Western blot bands and protein relative expression of AQP1, AQP2, AQP3 and AQP4 after compound **5** administration. The data are represented as the mean ± SD, *n* = 3, * *p* < 0.05 vs. control group (Mann–Whitney U test). Compound **2**: ent-11α-hydroxyabicta-8(14),13(15)-dien-16,12-olide; compound **3**: jolkinolide B; compound **4**: euphebracteolatin A; compound **5**: fischeria A.

**Figure 9 molecules-26-00942-f009:**
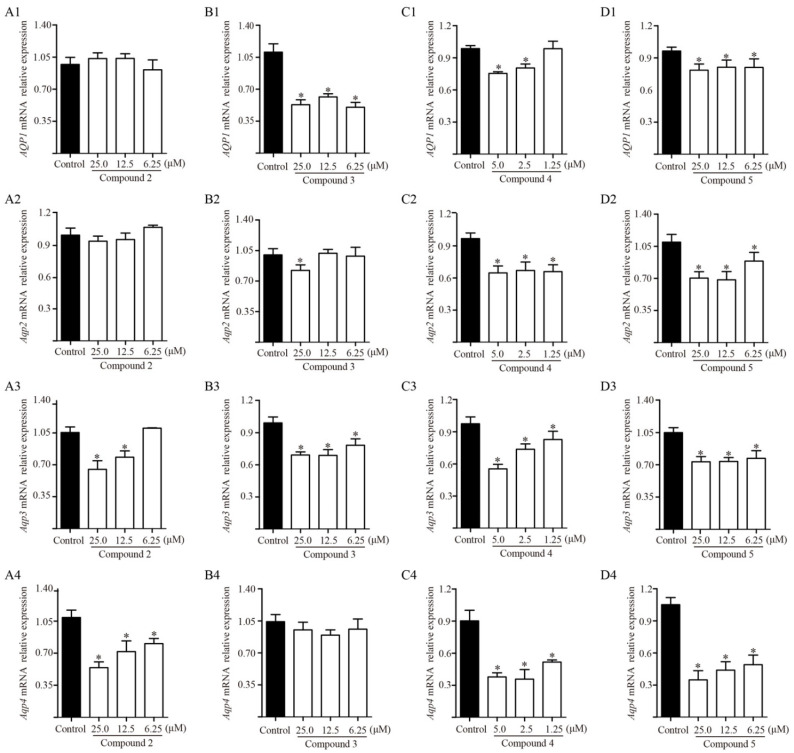
Effect of compounds **2**, **3**, **4** and **5** on mRNA expression levels of aquaporin (*AQP*)*-1* in human kidney-2 cells (HK-2), and *Aqp2*, *Aqp3* and *Aqp4* in murine inner medullary collecting duct cells (mIMCD3). (**A1**–**A4**) *AQP1*, *Aqp2*, *Aqp3* and *Aqp4* mRNA relative expression after compound **2** administration. (**B1**–**B4**) *AQP1*, *Aqp2*, *Aqp3* and *Aqp4* mRNA relative expression after compound **3** administration. (**C1**–**C4**) *AQP1*, *Aqp2*, *Aqp3* and *Aqp4* mRNA relative expression after compound **4** administration. (**D1**–**D4**) *AQP1*, *Aqp2*, *Aqp3* and *Aqp4* mRNA relative expression after compound **5** administration. The data are represented as the mean ± SD, *n* = 3, * *p* < 0.05 vs. control group (Mann–Whitney U test). Compound **2**: ent-11α-hydroxyabicta-8(14),13(15)-dien-16,12-olide; compound **3**: jolkinolide B; compound **4**: euphebracteolatin A; compound **5**: fischeria A.

**Figure 10 molecules-26-00942-f010:**
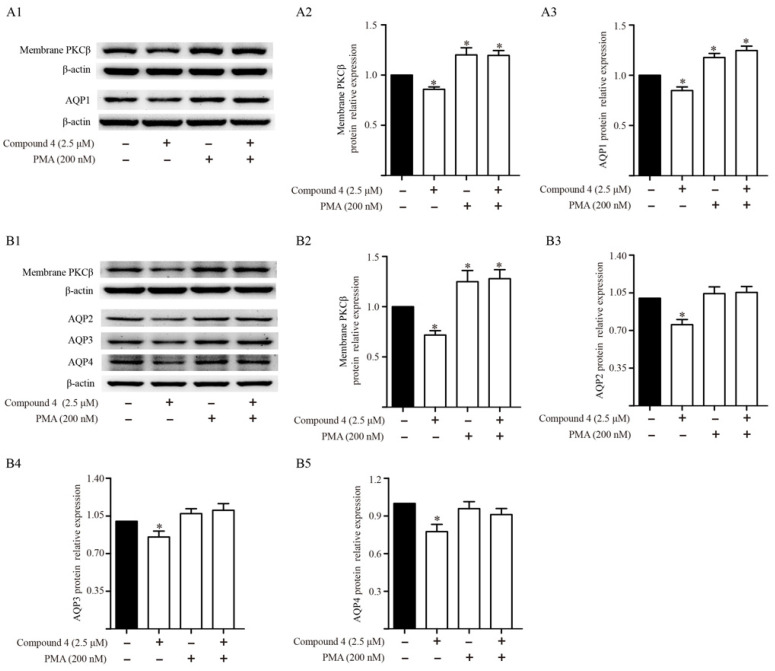
Effects of activator of protein kinase C (PKC) on diterpenoid-induced inhibition of aquaporin (AQP)-1 in human kidney-2 cells (HK-2), and AQP2, AQP3 and AQP4 in murine inner medullary collecting duct cells (mIMCD3). (**A1**–**A3**) Western blot bands and relative protein expression of membrane PKCβ and AQP1 in HK-2 cells. (**B1**–**B5**) Western blot bands and relative protein expression of membrane PKCβ, AQP2, AQP3 and AQP4 in mIMCD3 cells. The data are represented as the mean ± SD, *n* = 3, * *p* < 0.05 vs. control group (Mann–Whitney U test).

**Table 1 molecules-26-00942-t001:** mRNA primer sequences.

Gene	Forward (5′→3′)	Reverse (5′→3′)
Human *AQP1*	TGCCATCGGCCTCTCTGTAG	AAGGACCGAGCAGGGTTAATC
Mouse *Aqp2*	TGGCTGTCAATGCTCTCCAC	GGAGCAGCCGGTGAAATAGA
Mouse *Aqp3*	GAATCGTTGTGGGGAGATGC	CAAGATGCCAAGGGTGACAG
Mouse *Aqp4*	ATCAGCATCGCTAAGTCCGTC	GAGGTGTGACCAGGTAGAGGA
Human β-actin	GACCCAGATCATGTTTGAGAC	GTAGCCACGCTCGGTCAG
Mouse β-actin	TGGCTCCTAGCACCATGAAG	CCTGCTTGCTGATCCACATC

## Data Availability

The data presented in this study are available on request from the corresponding author.

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
