# Peer review of "Anti-Malignant Ascites Effect of Total Diterpenoids from Euphorbiae ebracteolatae Radix Is Attributable to Alterations of Aquaporins via Inhibiting PKC Activity in the Kidney"

_molecules, 2021, doi:10.3390/molecules26040942_

Round 1

Reviewer 1 Report

The manuscript from Yuanbin Zhang et al. describes the anti-ascites effect of the total diterpenoids extracted from Euphorbiae Ebracteolatae Radix on malignant ascitic mice and also confirms its underlying mechanism in human kidney-2 cells. The paper is well structured, and the experiments are well conducted. Some issues need to be fixed before acceptance. I have the following observations:

- The definition of “crude drug” is not appropriate, replace it with a more related term throughout the text

- Keywords: Replace the words already in the title

- Introduction: replace Euphorbiae Ebracteolatae Radix with the scientific name used in materials and methods

- Please always reports and specifies all the acronyms in the legends present in the figures and also describes all the different panels present (A, B, etc.)

- Please always reports and specifies all the acronyms in the legends present in the figures including the words "Normal" and "Model"

- Results section 2,4: please report more detailed results for AQPs expression and highlight the differences among TEER, TDEE and NTDEE treatments

- A double section 2.4 is present in the text

-  Section repeated 2.4: Why haven't the authors conducted the same tests with TEER since the dose of 0.6 appears to have the same effect as TDEE 3.0 on AQPs expression?

- In the legend of figure 6 for the panel B add the specification “standards”

- section 2.6 is missing

- Figure 7 panel A shows the images of the membranes of the western blot analysis, why do the authors not report the relative histograms but only the expression of the corresponding RNA? Please report the corresponding protein expression graphs!

- As in figure 7 also in figure 8 panel A1, B1, C1, D1 show the images of the membranes of the western blot analysis, why do the authors not report the relative histograms but only the expression of the corresponding RNA? Please report the corresponding protein expression graphs!

- Figure 9: Please report the corresponding protein expression graphs!

- Discussion “Previous studies have shown that EER and its diterpenoids have inhibitory activity on tumour cells in vitro” The sentence is not consistent with the bibliography reported. Please rephrase the sentence according to it.

- 4.4.4. Tumour cell viability, cycle, and apoptosis assays: Please describe in detail the method used to isolate the cells from the ascitic mice, and the detailed methods used for the tests presented in the section (number of cells etc.)

I would recommend the article could be published in Molecules, after a major revision. The authors need to address the below-mentioned queries.

Author Response

Answers to comments of Reviewer: 1
1. The definition of “crude drug” is not appropriate, replace it with a more related term throughout the text

Answer: Thank you very much for your reminding. We have replaced “crude drug” with “raw herbs” in the full text.

  1. Keywords: Replace the words already in the title.

Answer: Thank you very much. We have revised the Keywords to “Euphorbia ebracteolata Hayata; diuresis; bioactive components; terpenoids; kidney cells; PKC inhibitor”.

  1. Introduction: replace Euphorbiae Ebracteolatae Radix with the scientific name used in materials and methods

Answer: Thank you very much for your suggestions. We have replaced “Euphorbiae Ebracteolatae Radix” with “Euphorbia ebracteolata Hayata” in Introduction.

  1. Please always reports and specifies all the acronyms in the legends present in the figures and also describes all the different panels present (A, B, etc.)

Answer: Thank you very much for your suggestions. We have specified all the acronyms and described all the different panels in the figure legends.

  1. Please always reports and specifies all the acronyms in the legends present in the figures including the words "Normal" and "Model".

Answer: Thank you very much for your suggestions. We have specified all the acronyms in the figure legends.

  1. Results section 2.4: please report more detailed results for AQPs expression and highlight the differences among TEER, TDEE and NTDEE treatments.

Answer: Thank you very much for your suggestions. We have rewrote this section and added “The kidney is an important organ for water reabsorption and urinary concentration. AQP water channels mediate rapid transport of water in the kidneys. Thus, we examined the expression of AQPs in the kidneys. Figure 3 shows that the enhancement of ascites had little effects on AQPs expression in the kidneys in comparison with the normal group. When treated with TEER and TDEE, the expression of AQP1, AQP2, AQP3 and AQP4 proteins was markedly lower than in the model group at most doses. TEER resulted in a significant decrease in AQP2, AQP3 and AQP4 expression at doses of 3.0 and 0.6 g raw herbs/kg, whereas AQP1 expression was only reduced at low dose (0.6 g raw herbs/kg). Notably, TDEE had the same inhibitory effects on AQP3 and AQP4 expression as TEER at two doses. The difference between TDEE and TEER is that TDEE inhibited the expression of AQP1 and AQP2 only at high dose (3 g raw herbs/kg). NTDEE had week effects on AQPs levels in kidneys, only regulated the expression of AQP4 protein at a dose of 3 g raw herbs/kg.” in this part.

  1. A double section 2.4 is present in the text.

Answer: Thank you very much for your reminding. We have revised in the text.

  1. Section repeated 2.4: Why haven't the authors conducted the same tests with TEER since the dose of 0.6 appears to have the same effect as TDEE 3.0 on AQPs expression?

Answer: Thanks for your careful guidance. Our manuscript mainly wants to prove that diterpenoids are the main anti-ascites components of the root of Euphorbia ebracteolata Hayata (EER). Through the animal experiments, we confirmed that the total diterpenoids fraction (TDEE) is the effective fraction of EER in anti-ascites effect, so we directly used the total diterpenoids fraction to study the possible regulatory mechanism of AQPs in this section. Based on this, we continued to study the effect of the main components of TDEE.

  1. In the legend of figure 6 for the panel B add the specification “standards”

Answer: Thank you very much for your suggestions. We have added “standards” in the legend of figure 6.

  1. Section 2.6 is missing

Answer: We thank the reviewer for pointing out this error, we have now corrected this error.

  1. (1) Figure 7 panel A shows the images of the membranes of the western blot analysis, why do the authors not report the relative histograms but only the expression of the corresponding RNA? Please report the corresponding protein expression graphs!

(2) As in figure 7 also in figure 8 panel A1, B1, C1, D1 show the images of the membranes of the western blot analysis, why do the authors not report the relative histograms but only the expression of the corresponding RNA? Please report the corresponding protein expression graphs!

(3) Figure 9: Please report the corresponding protein expression graphs!

Answer: Thank you very much for your suggestions. We initially did not put the graphs of relative protein expression in the main article for reasons of length. Those graphs were added to corresponding position.

  1. Discussion “Previous studies have shown that EER and its diterpenoids have inhibitory activity on tumour cells in vitro” The sentence is not consistent with the bibliography reported. Please rephrase the sentence according to it.

Answer: Thanks for your reminding. We have changed this sentence to: “Previous studies have shown that the diterpenoids from EER have inhibitory activities on tumor cells in vitro”, and revised the references accordingly.

  1. 4.4.4. Tumor cell viability, cycle, and apoptosis assays: Please describe in detail the method used to isolate the cells from the ascitic mice, and the detailed methods used for the tests presented in the section (number of cells etc.)

Answer: Thank you very much for your suggestions. We have rewrote this section and added “After the mice were sacrificed, the ascitic fluid containing the tumor cells was collected from the peritoneal cavity and diluted 20 times with saline immediately. The viability of tumor cells was assessed by 0.4% trypan blue staining and counted in an automated cell counter (ALIT Life Science Co., Ltd., Shanghai, China). The diluted cells suspension (0.5 mL) was centrifuged at 1000 g for 5 min at 4 °C. The cells were collected and fixed in cold 70% ethanol at 4 °C overnight. After fixation, tumor cells were washed twice in PBS and centrifuged, and then incubated with RNase A solution (100 μL) for 30 min at 37 °C. Finally, cells were incubated with 400 μL propidium io-dide (PI) for 30 min at 4°C in the dark and filtered through a 300-mesh nylon membrane. The percentage of cells in different phase was analyzed using an Accuri C6 Flow Cytometer (BD Biosciences, San Jose, USA). Tumor cell apoptosis was detected using an Annexin V-FITC/PI Apoptosis kit. The diluted cells suspension (0.5 mL) was washed twice in PBS and centrifuged at 1000 g for 5 min at 4 °C. The cells were collected and resuspended in 100 uL binding buffer followed by incubation with 5 μL Annexin V-FITC and 10 μL PI staining solution for 15 min at room temperature in the dark. Following incubation, 400 μL binding buffer was added to each tube. The cell apoptosis was detected by using an Accuri C6 Flow Cytometer. ” in section 4.4.4..

Thank you very much for your attention.

Looking forward to hearing from you.

Yours sincerely,

Hongli Yu

E-mail: yuhongli76@163.com

Reviewer 2 Report

In this manuscript molecules-1097520, authors Zhang et al. study the mechanism for the anti-ascites effect observed for the diterpenoids extracted from the plant Euphorbiae bracteolatae radix (TDEE) on malignant ascetic mice.

The authors found that treatment of mice with TDEE caused reuction in ascites and improved urine output. They conclude, based on experiments performed, that TDEE treatment reduced the expression of AQP-1 as well as PKC-beta, and further propose that this is the underlying mechanism for the anti-ascitic effect observed with TDEE.

Overall, the study is sound in terms of experimental design and data presentation, as well as the conclusions drawn based on them.

A few minor points that I wish to list which the authors need to address:

1) The authors have used furosemide in their studies, I assume as a positive control. However, they have not mentioned or explained this anywhere in the text. They need to state this clearly either in methods or results (the purpose of using furosemide).

2) In figures 1, 3 and 4, p-value has been mentioned. However, the authors need to state which statistical test was performed, and what pair-wise comparisons the p-values represent.

Author Response

Answers to comments of Reviewer: 2
1. The authors have used furosemide in their studies, I assume as a positive control. However, they have not mentioned or explained this anywhere in the text. They need to state this clearly either in methods or results (the purpose of using furosemide).

 Answer: Thank you very much for your suggestions. We have added “Furosemide (6 mg/kg) was chosen as positive control because it is extensively used to treat edematous diseases such as ascites by diuresis effect.” in section 4.4.1. of Methods, and marked furosemide as a positive drug at it first appeared in Result.

  1. In figures 1, 3 and 4, p-value has been mentioned. However, the authors need to state which statistical test was performed, and what pair-wise comparisons the p-values represent.

Answer: We thank the reviewer for the suggestions, and we have added statistical test method at all the figure legends.

Thank you very much for your attention.

Looking forward to hearing from you.

Yours sincerely,

Hongli Yu

E-mail: yuhongli76@163.com

Round 2

Reviewer 1 Report

Authors adequately met the suggested indications and excellently reworked the manuscript. I recommend it for possible publication